# Retrieve, Reason, and Refine:
# Generating Accurate and Faithful Patient Instructions

Fenglin Liu[1]*, Bang Yang[2]*, Chenyu You[3], Xian Wu[4], Shen Ge[4], Zhangdaihong Liu[1,5]

Xu Sun[6]†, Yang Yang[7]†, David A. Clifton[1,5]

[1]Department of Engineering Science, University of Oxford  [2]School of ECE, Peking University
[3]Department of Electrical Engineering, Yale University  [4]Tencent JARVIS Lab, China
[5]Oxford-Suzhou Centre for Advanced Research, China
[6]MOE Key Lab of Computational Linguistics, School of Computer Science, Peking University
[7]School of Public Health, Shanghai Jiao Tong University School of Medicine, China
{fenglin.liu, david.clifton}@eng.ox.ac.uk, {bangyang, xusun}@pku.edu.cn
chenyu.you@yale.edu, jessie.liu@oxford-oscar.cn
{kevinxwu, shenge}@tencent.com, emma002@sjtu.edu.cn

## Abstract

The "Patient Instruction" (PI), which contains critical instructional information provided both to carers and to the patient at the time of discharge, is essential for the patient to manage their condition outside hospital. An accurate and easy-to-follow PI can improve the self-management of patients which can in turn reduce hospital readmission rates. However, writing an appropriate PI can be extremely time-consuming for physicians, and is subject to being incomplete or error-prone for (potentially overworked) physicians. Therefore, we propose a new task that can provide an objective means of avoiding incompleteness, while reducing clinical workload: the automatic generation of the PI, which is imagined as being a document that the clinician can review, modify, and approve as necessary (rather than taking the human "out of the loop"). We build a benchmark clinical dataset and propose the Re³Writer, which imitates the working patterns of physicians to first **re**trieve related working experience from historical PIs written by physicians, then **re**ason related medical knowledge. Finally, it **re**fines the retrieved working experience and reasoned medical knowledge to extract useful information, which is used to generate the PI for previously-unseen patient according to their health records during hospitalization. Our experiments show that, using our method, the performance of five different models can be substantially boosted across all metrics, with up to 20%, 11% and 19% relative improvements in BLEU-4, ROUGE-L and METEOR, respectively. Meanwhile, we show results from human evaluations to measure the effectiveness in terms of its usefulness for clinical practice. [3]

## 1   Introduction

At the time of discharge to home, the Patient Instruction (PI), which is a rich paragraph of text containing multiple instructions, is provided by the attending clinician to the patient or guardian. PI

---

*Equal contribution.
†Corresponding authors.
[3]The code is available at https://github.com/AI-in-Health/Patient-Instructions

36th Conference on Neural Information Processing Systems (NeurIPS 2022).

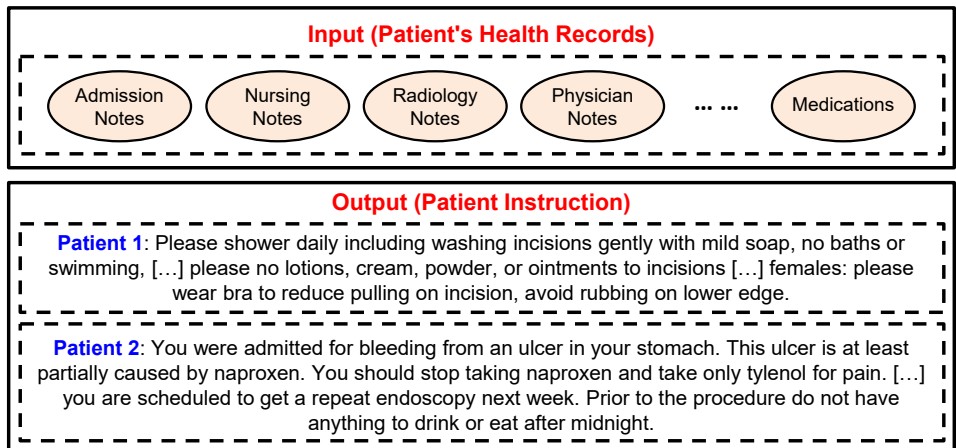

Figure 1: Two examples of the Patient Instruction written by physicians which guide the patients how to manage their conditions after discharge based on their health records during hospitalization.

is used for the purpose of facilitating safe and appropriate continuity of care [6, 37, 43]. As a result, the PI has significant implications for patient management and good medical care, lowering down the risks of hospital readmission, and improving the doctor-patient relationship. As shown in Figure 1, a PI typically contains the following three main components from patients' perspective [16, 45]: (1) What is my main health condition? (i.e., why was I in the hospital?) (2) What do I need to do? (i.e., how do I manage at home, and how should I best care for myself?) (3) Why is it important for me to do this? Of the above, the second section is often considered to be the most important information. For example, when a patient has had surgery while in the hospital, the PI might tell the patient to keep the wound away from water to avoid infection.

Currently, the following skills are needed for physicians to write a PI [37, 43, 6]: (1) thorough medical knowledge for interpreting the patient's clinical records, including diagnosis, medication, and procedure records; (2) skills for carefully analyzing the extensive and complex patient's data acquired during hospitalization, e.g., admission notes, nursing notes, radiology notes, physician notes, medications, and laboratory results; (3) the ability of extracting key information and writing instructions appropriate for the lay reader. Therefore, writing PIs is a necessary but time-consuming task for physicians, exacerbating the workload of clinicians who would otherwise focus on patient care [50, 41]. Besides, physicians need to read lots of patient's health records in their daily work, resulting in substantial opportunity for incompleteness or inappropriateness of wording [44, 51]. Statistically, countries with abundant healthcare resources, such as the United States, have up to 54% of physicians experiencing some sign of burnout in one year of study [42], which is further exacerbated in countries with more tightly resource-constrained healthcare resources.

The overloading of physicians is a well-documented phenomenon [44, 51], and AI-related support systems that can partly automate routine tasks, such as generation of PIs for modification/approval by clinicians is an important contribution to healthcare practice. To this end, we propose the novel task of automatic PI generation, which aims to generate an accurate and fluent textual PI given input health records of a previously-unseen patient during hospitalization. In this way, it is intended that physicians, given the health records of a new patient, need only review and modify the generated PI, rather than writing a new PI from scratch, significantly relieving the physicians from the heavy workload and increasing their time and energy spent in meaningful clinical interactions with patients. Such endeavors would be particularly useful in resource-limited countries [46].

In this paper, we build a dataset PI and propose a deep-learning approach named Re³Writer, which imitates the physicians' working patterns to automatically generate a PI at the point of discharge from hospital. Specifically, when a patient discharges from the hospital, physicians will carefully analyze the patient's health records in terms of diagnosis, medication, and procedure, then accurately write a corresponding PI based on their *working experience* and *medical knowledge* [5, 6]. In order to model clinicians' text production, the Re³Writer, which introduces three components: **Re**trieve, **Re**ason, and **Re**fine, (1) first encodes *working experience* by mining historical PIs, i.e., retrieving instructions

of previous patients according to the similarity of diagnosis, medication and procedure; (2) then reasons *medical knowledge* into the current input patient data by learning a knowledge graph, which is constructed to model domain-specific knowledge structure; (3) at last, refines relevant information from the retrieved working experience and reasoned medical knowledge to generate final PIs.

To prove the effectiveness of our Re$^3$Writer, we incorporate it into 5 different language generation models: 1) recurrent neural network (RNN)-based model, 2) attention-based model, 3) hierarchical RNN-based model, 4) copy mechanism-based model, 5) fully-attentive model, i.e., the Transformer [47]. The extensive experiments show that our approach can substantially boost the performance of baselines across all metrics.

Overall, the main contributions of this paper are:

- We make the first attempt to automatically generate the Patient Instruction (PI), which can reduce the workload of physicians. As a result, it can increase their time and energy spent in meaningful interactions with patients, providing high-quality care for patients and improving doctor-patient relationships.

- To address the task, we build a dataset PI and propose an approach Re$^3$Writer, which imitates physicians' working patterns to retrieve working experience and reason medical knowledge, and finally refine them to generate accurate and faithful patient instructions.

- We prove the effectiveness and the generalization capabilities of our approach on the built PI dataset. After including our approach, performances of the baseline models improve significantly on all metrics, with up to 20%, 11%, and 19% relative improvements in BLEU-4, ROUGE-L, and METEOR, respectively. Moreover, we conduct human evaluations to the generated PI for its quality and usefulness in clinical practice.

## 2 Related Works

The related works are introduced from two aspects: 1) Natural Language Generation (NLG) and 2) Medical Text Generation.

**Natural Language Generation (NLG)**   It aims to automatically generate coherent text in natural-language from given corresponding input data, which can be, e.g., text [1], images [49], video [48], and audio [8]. Commonly, approaches use an encoder-decoder framework, where the encoder computes intermediate representations for the input source data and the decoder adopts RNNs or CNNs [14, 9], to generate the target sentences given the intermediate representation. The attention mechanism [33, 47, 52, 27] and the copy mechanism [11] have been proposed to directly provide the decoder with the source information, enabling a more efficient use of the source data. In particular, fully-attentive models, such as the Transformer [47], in which no recurrence is required, have successfully achieved state-of-the-art performance for multiple natural language generation tasks. However, most existing work focuses on the natural text-related data, where research concerning the clinical domain remains relatively under-studied.

**Medical Text Generation**   Recently, language generation in the medical domain has received growing research interest. For example, Jing et al. [18], Liu et al. [29, 30] proposed to generate radiology reports from chest X-ray images; Lee [23] proposed to generate the clinical notes of emergency department cases from discharge diagnosis codes; Ive et al. [17] proposed to generate mental health records. Meanwhile, some works treated the generation of medical text as a summarization task. For example, Zhang et al. [53] proposed to generate clinical impressions by summarizing radiology reports; Scott et al. [39] proposed to generate the patients' histories by summarizing their respective health records; Several works [36, 12, 32, 10, 40], including HARVEST [13] - a medical summarization tool, proposed to summarize patients' key conditions from their clinical health records, such as Hassanpour and Langlotz [12] and Shing et al. [40] which summarized important clinical entities from radiology reports and hospital health records, respectively.

Despite the encouraging performances of these methods in generating various types of medical text, to the best of our knowledge, none of them has attempted to generate the Patient Instructions, which has significant implications for supporting care delivery [6, 37, 43] as noted earlier. We here make the first attempt to generate the Patient Instructions, which is typically longer than existing NLG-related output in the medical domain and which covers more diverse topics than previous medical text generation tasks, such as clinical impressions [53], clinical notes [23], and radiology reports [28, 31].

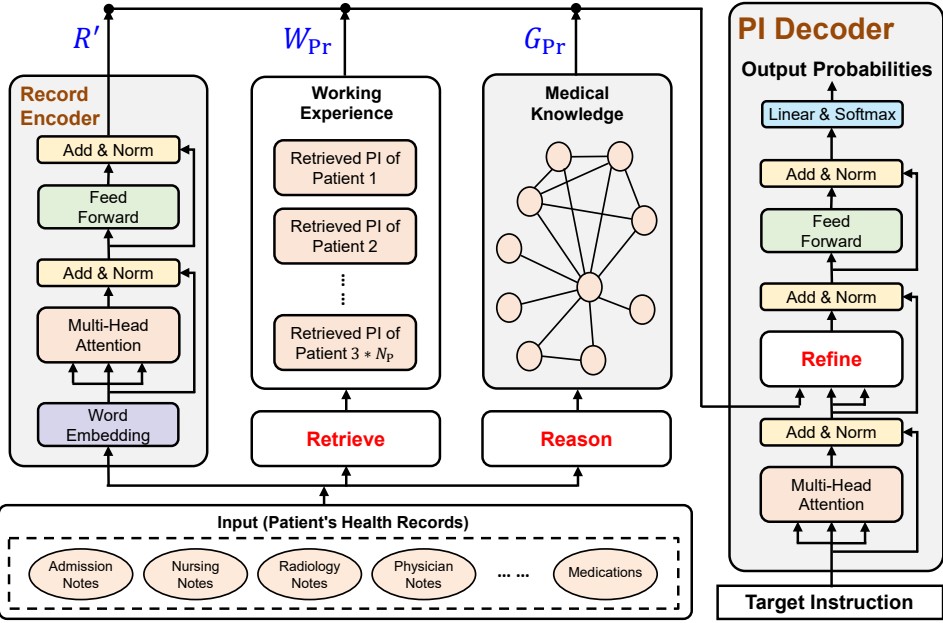

Figure 2: We take the Transformer [47] as our baseline as an example to illustrate our Re³Writer: Retrieve, Reason, and Refine, which is designed to first retrieve related working experience from historical PIs written by physicians and reason related medical knowledge from a medical knowledge graph; then adaptively refine and merge them to generate accurate and faithful patient instruction for current previously-unseen patient.

## 3 Approach

We first define the PI generation problem; then, we describe the proposed Re³Writer in detail.

### 3.1 PI Generation Problem Definition

When a patient is discharged from the hospital, the PI generation system should generate a fluent and faithful instruction to help the patient or carer to manage their conditions at home. Therefore, the goal of the PI generation task is to generate a target instruction $I = \{y_1, y_2, \ldots, y_{N_I}\}$ given the patient's health records $R = \{r_1, r_2, \ldots, r_{N_R}\}$ in terms of diagnoses, medications and procedures performed during hospitalization.

Since the input $R$ including $N_R$ words and the output $I$ including $N_I$ words are both textual sequences, we adopt the encoder-decoder framework, which is widely-used in natural language generation tasks, to perform the PI generation task. In particular, the encoder-decoder framework includes a health record encoder and a PI decoder, which can be formulated as:

$$\text{Record Encoder}: R \to R'; \ \text{PI Decoder}: R' \to I, \tag{1}$$

where $R' \in \mathbb{R}^{N_R \times d}$ denotes the record embeddings encoded by the record encoder, e.g., LSTM [14] or Transformer [47]. Then, $R'$ is fed into the PI decoder (which again could be an LSTM or Transformer), to generate the target PI $I$. During training, given the ground truth PI for the input patient's health records, we can train the model by minimizing the widely-used cross-entropy loss.

### 3.2 The Proposed Re³Writer

Our Re³Writer consists of three core components: Retrieve, Reason, and Refine.

**Formulation of the Re³Writer**  As stated above, given the health records $R$ encoded as $R' \in \mathbb{R}^{N_R \times d}$, we aim to generate a desirable PI $I$. Figure 2 shows the detail of our method, which is designed to retrieve related working experience $W_{Pr}$ and reason related medical knowledge $G_{Pr}$ from the training corpus for current input patient. Finally, Re³Writer refines the retrieved working

experience $W_\mathrm{Pr}$ and reasoned medical knowledge $G_\mathrm{Pr}$ to extract useful information to generate a proper PI:

$$\text{Record Encoder} : R \to R'; \ \text{Retrieve} : R \to W_\mathrm{Pr}; \ \text{Reason} : R \to G_\mathrm{Pr};$$
$$\text{Refine + PI Decoder} : \{R', W_\mathrm{Pr}, G_\mathrm{Pr}\} \to I. \tag{2}$$

Our method can be incorporated into existing encoder-decoder based models, to boost their performance with PI generation, as we will later demonstrate. We now describe how to retrieve related working experience and reason related medical knowledge from the training corpus for PI generation.

**Retrieve**   As shown in Figure 1, a hospitalization typically produces records of diagnoses given, medications used, and procedures performed; our dataset (described later) has 11,208 unique clinical codes, including 5,973 diagnosis codes, 3,435 medication codes, and 1,800 procedure codes. Therefore, we first extract the one-hot embeddings of all clinical codes. Given a new patient, we represent this patient's hospitalization by averaging the associated one-hot embeddings of clinical codes produced during this hospitalization. Then, we collect a set of patients similar to the new patient according to the associated clinical codes. Taking the diagnosis codes as an example, we retrieve $N_\mathrm{P}$ patients from the training corpus with the highest cosine similarity to the input diagnosis codes of the current patient. The PIs of the top-$N_\mathrm{P}$ retrieved patients are returned and encoded by a BERT encoder [7, 38], followed by a max-pooling layer over all output vectors, and projected to $d$ dimensions, generating the related working experience in terms of Diagnosis for the current patient: $\{I_1, I_2, \dots, I_{N_\mathrm{P}}\} \in \mathbb{R}^{N_\mathrm{P} \times d}$. Similarly, we use the medication codes and procedure codes to acquire the working experience in terms of Medication and Procedure, respectively. At last, we concatenate these code-specific working experience representation (Diagnosis, Medication, Procedure) to obtain the final working experience related to current patient $W_\mathrm{Pr} = \{I_1, I_2, \dots, I_{3*N_\mathrm{P}}\} \in \mathbb{R}^{(3*N_\mathrm{P}) \times d}$. We also attempted to incorporate age and gender information into our approach to match patients, please see Appendix C for details.

**Reason**   To reason related medical knowledge from the training corpus, we construct an off-the-shelf global medical knowledge graph $\mathcal{G} = (V, E)$ using all clinical codes, i.e., diagnosis, medication, and procedure codes, across all hospitalizations, where $V = \{v_i\}_{i=1:N_\mathrm{KG}} \in \mathbb{R}^{N_\mathrm{KG} \times d}$ is a set of $N_\mathrm{KG}$ nodes and $E = \{e_{i,j}\}_{i,j=1:N_\mathrm{KG}}$ is a set of edges. The $\mathcal{G}$ models the domain-specific knowledge structure. In detail, we consider the clinical codes as nodes. The weights of the edges are calculated by normalizing the co-occurrence of pairs of nodes in the training corpus. After that, guided by current input patient's health records, the knowledge graph is embedded by a graph convolution network (GCN) [34, 24, 22], acquiring a set of node embeddings $\{v'_1, v'_2, \dots, v'_{N_\mathrm{KG}}\}$, which is regarded as our reasoned medical knowledge $G_\mathrm{Pr} = \{v'_1, v'_2, \dots, v'_{N_\mathrm{KG}}\} \in \mathbb{R}^{N_\mathrm{KG} \times d}$.

Please refer to Appendix A for the detailed description of our medical knowledge graph. We note that more complex graph structures could be constructed by using larger-scale well-defined medical ontologies. Therefore, our approach is not limited to the currently constructed graph and could provide a good basis for future research in this direction.

**Refine**   As shown in Eq. (2), the PI decoder equipped with our method aims to generate the final PI based on the encoded patient's health records $R' \in \mathbb{R}^{N_\mathrm{R} \times d}$, the retrieved working experience $W_\mathrm{Pr} \in \mathbb{R}^{(3*N_\mathrm{P}) \times d}$ and the reasoned medical knowledge $G_\mathrm{Pr} \in \mathbb{R}^{N_\mathrm{KG} \times d}$. In implementations, we can choose either the LSTM [14] or Transformer [47] as the decoder. Taking the Transformer decoder as an example: for each decoding step $t$, the decoder takes the embedding of the current input word $x_t = w_t + e_t \in \mathbb{R}^d$ as input, where $w_t$ and $e_t$ denote the word embedding and fixed position embedding, respectively; we then generate each word $y_t$ in the target instruction $I = \{y_1, y_2, \dots, y_{N_\mathrm{I}}\}$:

$$h_t = \mathrm{MHA}(x_t, x_{1:t}); \ h'_t = \mathrm{Refine}(h_t, R', W_\mathrm{Pr}, G_\mathrm{Pr}); \ y_t \sim p_t = \mathrm{softmax}(\mathrm{FFN}(h'_t)\mathrm{W}_p + \mathrm{b}_p) \tag{3}$$

where the MHA and FFN respectively stand for the Multi-Head Attention and Feed-Forward Network in the original Transformer (see Appendix D); $\mathrm{W}_p \in \mathbb{R}^{d \times |D|}$ and $\mathrm{b}_p$ are the learnable parameters ($|D|$: vocabulary size); the Refine component then refines the $W_\mathrm{Pr}$ and $G_\mathrm{Pr}$ to extract useful and correlated information to generate an accurate and faithful PI.

Intuitively, the PI generation task aims to produce an instruction based on the source patient's health records $R'$, supported with appropriate working experience $W_\mathrm{Pr}$ and medical knowledge $G_\mathrm{Pr}$. Thus, $W_\mathrm{Pr}$ and $G_\mathrm{Pr}$ play an auxiliary role during the PI generation. To this end, the Refine component,

which makes the model adaptively learn to refine correlated information, is designed as follows:

$$\text{Refine}(h_t, R', W_{\text{Pr}}, G_{\text{Pr}}) = \text{MHA}(h_t, R') + \lambda_1 \odot \text{MHA}(h_t, W_{\text{Pr}}) + \lambda_2 \odot \text{MHA}(h_t, G_{\text{Pr}})$$
$$\lambda_1 = \sigma\left([h_t; \text{MHA}(h_t, W_{\text{Pr}})]\mathbf{W}_{h1} + \mathbf{b}_{h1}\right); \quad \lambda_2 = \sigma\left([h_t; \text{MHA}(h_t, G_{\text{Pr}})]\mathbf{W}_{h2} + \mathbf{b}_{h2}\right) \quad (4)$$

where $\mathbf{W}_{h1}, \mathbf{W}_{h2} \in \mathbb{R}^{2d \times d}$ and $\mathbf{b}_{h1}, \mathbf{b}_{h2}$ are learnable parameters. $\odot$, $\sigma$, and $[\cdot; \cdot]$ denote the element-wise multiplication, the sigmoid function, and the concatenation operation, respectively. The computed $\lambda_1, \lambda_2 \in [0, 1]$ weight the expected importance of $W_{\text{Pr}}$ and $G_{\text{Pr}}$ for each target word.

In particular, if LSTM is adopted as the PI decoder, we can directly replace the MHA with the LSTM unit and remove the FFN in Eq. (3). In the following sections, we will prove that our proposed Re³Writer can provide a solid basis for patient instruction generation.

# 4  Experiment

We first introduce our dataset used as the basis for our experiments, as well as the baseline models and settings. We subsequently show evaluations using both automatic and "human" approaches.

## 4.1  Dataset, Baselines, and Settings

**Dataset**  We propose a benchmark clinical dataset Patient Instruction (PI) with around 35k pairs of input patient's health records and output patient instructions. In detail, we collect the PI dataset from the publicly-accessible MIMIC-III v1.4 resource[4][20, 19], which integrates de-identified, comprehensive clinical data for patients admitted to the Intensive Care Unit of the Beth Israel Deaconess Medical enter in Boston, Massachusetts, USA. This resource is an important "benchmark" dataset that promotes easy comparison between studies in the area. For each patient in the MIMIC-III v1.4 resource, the dataset includes various patient's health records during hospitalization in terms of diagnoses, medications and procedures, e.g., demographics, laboratory results, admission notes, nursing notes, radiology notes, and physician notes. In our experiments, we found that the discharge summaries contain the abstractive information of patient's health records. Therefore, for clarity, we adopt such abstractive information to generate the PI. We concatenate all available patients' health records as the input of our model.

For data preparation, we first exclude entries without a patient instruction and entries where the word-counts of patient instructions are less than five. This results in our PI dataset of 28,029 unique patients and 35,851 pairs of health records and patient instructions, as summarized in Table 1. We randomly partition the dataset into 80%-10%-10% train-validation-test partitions according to patients. Therefore, there is no overlap of patients between train, validation and test sets. Next, we pre-process the records

Table 1: Statistics of the built Patient Instruction (PI) dataset.

| Statistics | TRAIN | VAL | TEST |
|---|---|---|---|
| Number of Instructions | 28,673 | 3,557 | 3,621 |
| Number of Patients | 22,423 | 2,803 | 2,803 |
| Avg. Instruction Length | 162.5 | 164.5 | 162.8 |
| Avg. Record Length | 2147.1 | 2144.9 | 2124.3 |

and instructions by tokenizing and converting text to lower-case. Finally, we filter tokens that occur fewer than 20 times in the corpus, resulting in a vocabulary of approximately 19.9k tokens, which covers over 99.5% word occurrences in the dataset.

**Baselines**  We choose five representative language generation models with different structures as baseline models, i.e., 1) RNN-based model (**LSTM**) [4], 2) attention-based model (**Seq2Seq**) [1, 33], 3) hierarchical RNN-based model (**HRNN**) [26], 4) copy mechanism based model (**CopyNet**) [11], 5) fully-attentive model (**Transformer**) [47]. We prove the effectiveness of our Re³Writer by comparing the performance of the various baseline models with and without the Re³Writer.

**Settings**  The model size $d$ is set to 512. For a patient, we directly concatenate all available patient's health records during hospitalization as input, e.g., admission notes, nursing notes, radiology notes. For example, if a patient only has admission notes, our model will just rely on the available admission notes to generate the PI. We adopt Transformer [47] as the record encoder. Based on the average performance on the validation set, the number of retrieved previous PIs, $N_{\text{P}}$, is set to 20 for all three codes (see Appendix B). During training, we use the Adam optimizer [21] with a batch size of 128 and a learning rate of $10^{-4}$ for parameter optimization. We perform early stopping based on BLEU-4 with a maximum 150 epochs. During testing, we apply beam search of size 2 and a repetition penalty

---
[4]https://physionet.org/content/mimiciii/

Table 2: Performance of automatic evaluation on our built benchmark dataset PI. Higher is better in all columns. We conducted 5 runs with different seeds for all experiments, the t-tests indicate that $p < 0.01$. The (+Number) denotes the absolute improvements. As we can see, all the baseline models with significantly different structures enjoy a comfortable improvement with our Re³Writer approach.

| Methods | Dataset: Patient Instruction (PI) | | | | | | | |
|---|---|---|---|---|---|---|---|---|
| | METEOR | ROUGE-1 | ROUGE-2 | ROUGE-L | BLEU-1 | BLEU-2 | BLEU-3 | BLEU-4 |
| LSTM [4] | 16.5 | 35.9 | 17.9 | 33.2 | 34.4 | 26.3 | 23.1 | 21.0 |
| with Re³Writer | **19.6 (+3.1)** | **39.4 (+3.5)** | **20.5 (+2.6)** | **37.0 (+3.8)** | **40.8 (+6.4)** | **31.5 (+5.2)** | **27.6 (+4.5)** | **25.3 (+4.3)** |
| Seq2Seq [1] | 19.9 | 39.0 | 20.3 | 37.1 | 41.6 | 32.5 | 27.9 | 25.1 |
| with Re³Writer | **20.9 (+1.0)** | **40.8 (+1.8)** | **21.9 (+1.6)** | **38.6 (+1.5)** | **43.2 (+1.6)** | **34.2 (+1.7)** | **29.7 (+1.8)** | **26.8 (+1.7)** |
| HRNN [26] | 20.3 | 40.1 | 20.5 | 36.9 | 43.5 | 33.7 | 28.8 | 25.6 |
| with Re³Writer | **21.6 (+1.3)** | **42.5 (+2.4)** | **22.1 (+1.6)** | **39.0 (+2.1)** | **47.2 (+3.7)** | **36.9 (+3.2)** | **31.5 (+2.7)** | **27.8 (+2.2)** |
| CopyNet [11] | 19.5 | 38.3 | 19.9 | 36.5 | 40.4 | 31.6 | 27.0 | 24.4 |
| with Re³Writer | **20.6 (+1.1)** | **39.9 (+1.6)** | **20.9 (+1.0)** | **37.8 (+1.3)** | **42.7 (+2.3)** | **33.6 (+2.0)** | **28.7 (+1.7)** | **26.0 (+1.6)** |
| Transformer [47] | 21.8 | 42.1 | 21.6 | 38.9 | 47.1 | 36.8 | 31.4 | 27.3 |
| with Re³Writer | **23.7 (+1.9)** | **45.8 (+3.7)** | **24.4 (+2.8)** | **42.2 (+3.3)** | **52.4 (+5.3)** | **41.2 (+4.4)** | **35.0 (+3.6)** | **30.5 (+3.2)** |

Table 3: Performance of human evaluation for comparing our method (baselines with Re³Writer) with baselines in terms of the *fluency* of generated PIs, the *comprehensiveness* of the generated true PIs and the *faithfulness* to the ground truth PIs. All values are reported in percentage (%).

| Metrics | Seq2Seq [1] | | | Transformer [47] | | |
|---|---|---|---|---|---|---|
| | Baseline Win | Tie | Ours Win | Baseline Win | Tie | Ours Win |
| Fluency | 10.5 | 69.5 | **20.0** | 27.0 | 40.5 | **32.5** |
| Faithfulness | 22.5 | 38.0 | **39.5** | 25.5 | 31.0 | **43.5** |
| Comprehensiveness | 17.5 | 48.0 | **34.5** | 21.0 | 24.0 | **55.0** |

Table 4: Evaluation of how many times physicians would have deemed the generated result as "helpful" vs. "unhelpful" in terms of assisting them in writing PIs. Baseline denotes the Transformer [47].

| Methods | Helpful ↑ | Unhelpful ↓ |
|---|---|---|
| Baseline | 32% | 68% |
| Ours | **74%** | **26%** |

of 2.5. For all baselines, we keep the inner structure of baselines untouched and preserve the same training/testing settings for experiments.

## 4.2 Automatic Evaluation

**Metrics** We measure the performance by adopting widely-used natural language generation metrics, i.e., BLEU-1, -2, -3, -4 [35], METEOR [2] and ROUGE-1, -2, -L [25], which are calculated by the evaluation toolkit [3] and measure the match between the generated instructions and reference instructions annotated by professional clinicians.

**Results** Table 2 shows that our Re³Writer can consistently boost all baselines across all metrics, with a relative improvement of 7%~20%, 4%~11%, and 5%~19% in BLEU-4, ROUGE-L, and METEOR, respectively. The improved performance proves the validity of our approach in retrieving working experience, reasoning medical knowledge, and refining them for PI generation. The performance gains over all of the five baseline models also indicate that our approach is less prone to the variations of model structures and hyper-parameters, proving that the generalization capabilities of our approach are robust over a wide range of models. In Section 4.5, we verify the robustness of our approach to various data/examples.

## 4.3 Human Evaluation

**Metrics** We further conduct human evaluation to verify the effectiveness of our approach Re³Writer in clinical practice. To successfully assist physicians and reduce their workloads, it is important to generate accurate patient instructions (*faithfulness*, precision), such that the model does not generate instructions that "do not exist" according to doctors. It is also necessary to provide comprehensive true instructions (*comprehensiveness*, recall), i.e., the model does not leave out the important instructions. For example, given the ground truth PI [Instruction_A, Instruction_B] written by physicians, the PI generated by Model_1 is [Instruction_A, Instruction_B, Instruction_C], and the PI generated by

Table 5: Ablation study of our Re$^3$Writer, which includes three components: Retrieve, Reason, and Refine, on two representative baseline models, i.e., Seq2Seq [1] and Transformer [47]. Full Model denotes the baseline model with our Re$^3$Writer.

| Settings | Retrieve: Working Experience | | | Reason: Knowledge | Refine | Dataset: Patient Instruction (PI) | | | | | |
|---|---|---|---|---|---|---|---|---|---|---|---|
| | Diagnose | Medication | Procedure | | | METEOR | ROUGE-1 | ROUGE-2 | ROUGE-L | BLEU-3 | BLEU-4 |
| Seq2Seq | | | | | | 19.9 | 39.0 | 20.3 | 37.1 | 27.9 | 25.1 |
| (a) | ✓ | | | | | 20.1 | 39.1 | 20.5 | 37.2 | 28.3 | 25.5 |
| (b) | | ✓ | | | | 20.7 | 39.8 | 21.2 | 37.8 | 29.0 | 26.1 |
| (c) | | | ✓ | | | 20.1 | 39.2 | 20.6 | 37.2 | 28.4 | 25.6 |
| (d) | ✓ | ✓ | ✓ | | | 20.6 | 40.1 | 21.1 | 38.0 | 29.2 | 26.3 |
| (e) | | | | ✓ | | 20.7 | 40.5 | 21.6 | 38.4 | 29.4 | 26.5 |
| (f) | ✓ | ✓ | ✓ | ✓ | | 20.7 | 40.5 | 21.7 | 38.2 | 29.5 | 26.6 |
| Full Model | ✓ | ✓ | ✓ | ✓ | ✓ | **20.9** | **40.8** | **21.9** | **38.6** | **29.7** | **26.8** |
| Transformer | | | | | | 21.8 | 42.1 | 21.6 | 38.9 | 31.4 | 27.3 |
| (a) | ✓ | | | | | 22.2 | 43.1 | 22.3 | 39.7 | 32.3 | 28.0 |
| (b) | | ✓ | | | | 22.7 | 44.2 | 23.0 | 40.7 | 33.4 | 28.9 |
| (c) | | | ✓ | | | 22.5 | 43.6 | 22.7 | 40.1 | 32.7 | 28.4 |
| (d) | ✓ | ✓ | ✓ | | | 23.1 | 44.5 | 23.6 | 41.0 | 33.6 | 29.2 |
| (e) | | | | ✓ | | 23.2 | 44.8 | 23.8 | 41.4 | 34.0 | 29.4 |
| (f) | ✓ | ✓ | ✓ | ✓ | | 23.4 | 45.2 | 24.1 | 41.8 | 34.3 | 29.9 |
| Full Model | ✓ | ✓ | ✓ | ✓ | ✓ | **23.7** | **45.8** | **24.4** | **42.2** | **35.0** | **30.5** |

Model_2 is [Instruction_A]. Model_1 is better than Model_2 in terms of *comprehensiveness*, while is worse than Model_2 in terms of *faithfulness*. Finally, it is unacceptable to generate repeated or otherwise unreadable instructions (*fluency*). Therefore, we randomly select 200 samples from our PI dataset. The human evaluation is conducted by two junior annotators (medical students) and a senior annotator (clinician). All three annotators have sufficient medical knowledge. By giving the ground-truth PIs, each junior annotator was asked to independently compare the outputs of our approach and that of the baseline models, in terms of the perceived quality of the outputs - including *fluency*, *comprehensiveness*, and *faithfulness* of outputs compared to the corresponding ground-truth instructions. The senior annotator re-evaluated those cases that junior annotators have difficulties deciding. The annotators were blinded to the model that generated the output instructions.

**Results** We select a representative baseline, Seq2Seq, and a competitive baseline, Transformer, to show the human evaluation results (Table 3). As may be seen from the results, our Re$^3$Writer is better than baselines with improved performance in terms of the three aspects, i.e., *fluency*, *comprehensiveness* and *faithfulness*. The human evaluation results show that the instructions generated by our approach are of higher clinical quality than the competitive baselines, which proves the advantage of our approach in clinical practice. In particular, by using our Re$^3$Writer, the winning chances increased by a maximum of $43.5 - 25.5 = 18$ points and $55 - 21 = 34$ points in terms of the *faithfulness* metric (precision) and *comprehensiveness* metric (recall), respectively. At last, we further evaluate how many times physicians would have deemed the generated result as "helpful" vs. "unhelpful" in terms of assisting them in writing a PI. The results are reported in Table 4. As we can see, our approach can generate more accurate PIs than the baselines, improving the usefulness of AI systems in better assisting physicians in clinical decision-makings and reducing their workload.

## 4.4 Ablation Study

We conduct a quantitative analysis using the Seq2Seq and the Transformer for the purposes of evaluating the contribution of each proposed component: Retrieve, Reason, and Refine.

**Effect of the 'Retrieve' Component** Table 5 (a,b,c) shows that the Diagnosis, Medication, and Procedure elements of the model all contribute to a boost in performance, which proves the effectiveness of our approach in retrieving similar patient instructions from the available repository of historical PIs to aid in the generation of new patient instructions. Among these three elements, Medication leads to the best improvements, which may be explained by the fact that the PI is more relevant to medications from historical experience [45]. By combining the three elements (d), we observe an overall improvement. As a result, retrieving related working experience can boost the performance of baselines: $25.1 \rightarrow 26.3$ and $27.3 \rightarrow 29.2$ in BLEU-4 for Seq2Seq and Transformer, respectively.

Table 6: Performance of our approach on the three sub-datasets: Gender, Age, Disease.

| Gender | Female | | | Male | | |
|---|---|---|---|---|---|---|
| | METEOR | ROUGE-L | BLEU-4 | METEOR | ROUGE-L | BLEU-4 |
| Seq2Seq | 19.8 | 35.9 | 25.0 | 20.0 | 38.0 | 25.2 |
| with Re³Writer | **20.6** | **37.6** | **26.3** | **21.1** | **39.5** | **27.2** |
| Transformer | 21.5 | 38.1 | 26.9 | 22.0 | 39.6 | 27.6 |
| with Re³Writer | **23.2** | **41.3** | **30.1** | **24.1** | **43.0** | **30.8** |

| Age Group | Age<55 | | | 55<=Age<70 | | | Age>=70 | | |
|---|---|---|---|---|---|---|---|---|---|
| | METEOR | ROUGE-L | BLEU-4 | METEOR | ROUGE-L | BLEU-4 | METEOR | ROUGE-L | BLEU-4 |
| Seq2Seq | 18.2 | 34.7 | 21.9 | 20.7 | 39.5 | 26.7 | 20.7 | 37.1 | 26.5 |
| with Re³Writer | **19.2** | **35.6** | **23.7** | **21.8** | **41.2** | **28.4** | **21.5** | **38.9** | **28.1** |
| Transformer | 20.2 | 36.9 | 24.4 | 23.1 | 41.3 | 28.5 | 22.8 | 39.0 | 28.4 |
| with Re³Writer | **22.9** | **40.1** | **28.5** | **26.2** | **45.0** | **31.8** | **24.3** | **42.7** | **31.2** |

| Disease | Hypertension | | | Hyperlipidemia | | | Anemia | | |
|---|---|---|---|---|---|---|---|---|---|
| | METEOR | ROUGE-L | BLEU-4 | METEOR | ROUGE-L | BLEU-4 | METEOR | ROUGE-L | BLEU-4 |
| Seq2Seq | 21.3 | 39.8 | 27.9 | 21.3 | 41.7 | 27.2 | 18.0 | 36.4 | 20.7 |
| with Re³Writer | **22.6** | **41.4** | **30.4** | **22.5** | **43.9** | **29.5** | **18.8** | **37.6** | **22.3** |
| Transformer | 22.8 | 42.5 | 30.7 | 23.0 | 44.7 | 30.3 | 19.6 | 38.2 | 23.4 |
| with Re³Writer | **24.6** | **45.1** | **33.5** | **24.9** | **46.4** | **33.8** | **21.8** | **41.3** | **27.4** |

**Effect of the 'Reason' Component**   Table 5 (e) shows that the Reason component can further improve the performance by learning the enriched medical knowledge. By comparing (d) and (e), we can observe that the reasoned medical knowledge leads to similar improvements as the retrieved working experience does. We attribute this to the fact that learning conventional and general writing styles for PIs in a deep learning-based approach is as important as incorporating accurate medical knowledge. In this way, the retrieved patient instructions of previous patients can be treated as templates, providing a solid basis to generate accurate PIs.

**Effect of the 'Refine' Component**   As shown in Table 5 (f and Full Model), it is clear that the model with our Refine component performs better than the model without it, which directly verifies the effectiveness of our approach. To further understand the ability of adaptively refining useful information, we summarize the average refining weight values $\bar{\lambda}_1$ and $\bar{\lambda}_2$ of Eq. (4). We can find that the average value of $\bar{\lambda}_2 = 0.42$ is larger than $\bar{\lambda}_1 = 0.26$. It indicates that the medical knowledge plays a prominent role in PI generation; it is in accordance with the results of (d) and (e). We conclude that our method is capable of learning to efficiently refine the most related useful information to generate accurate PIs. Overall, our three components improve the performance from different perspectives, and combining them can lead to an overall improvement.

## 4.5   Robustness Analysis

In this section, we evaluate the performance of our approach with more fine-grained datasets. Specifically, we further divide our PI dataset into three sub-datasets according to Gender, Age, and Disease. Specifically, to ensure an even distribution of the data, we divide the ages into three age-groups: Age < 55 (29.9%), 55 <= Age < 70 (30.5%), and Age >= 70 (39.7%). Table 6 shows the results of our approach on the three sub-datasets. As we can see, the proposed approach can consistently boost baselines across different genders, ages, and diseases on all evaluation metrics, proving the generalization capability and the effectiveness of our method to different datasets/examples.

## 4.6   Qualitative Analysis

Figure 3 shows that our approach is significantly better aligned with ground truth instructions than the baseline. For example, our approach correctly generates the key instructions ("*you were found to have an infection in your gallbladder*" and "*you also had a urinary tract infection which was treated with antibiotics*" (Blue-colored text)), and the personalised medication note ( "*start oxycodone 5 mg*

<table>
<tr><td>

**Ground Truth:** dear ms. [name], you were admitted to the hospital with abdominal pain, nausea, vomiting and inability to eat. you were diagnosed with a urinary tract infection, and treated with antibiotics. your pain did not improve, and you were diagnosed with chronic cholecystitis (inflammation of your gallbladder). your gallbladder was removed. changes to your home medications include: for pain, you can take acetaminophen. you can also take oxycodone 5 - 10mg every 4 - 6 hours for pain. your pain should be improving, so you should require less medication over time.

</td></tr>
</table>

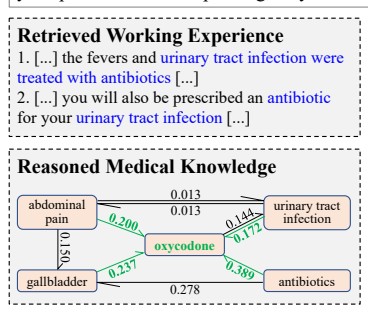

**Retrieved Working Experience**
1. [...] the fevers and urinary tract infection were treated with antibiotics [...]
2. [...] you will also be prescribed an antibiotic for your urinary tract infection [...]

**Reasoned Medical Knowledge**

**Baseline:** dear ms. [name], it was a pleasure taking care of you at [hospital]. you were admitted to the hospital for abdominal pain and nausea / vomiting. you were found to have an infection in your blood stream which we treated with antibiotics. you were also started on iv antibiotics for this infection. you will need to follow - up with dr. [name] as listed below.

**with Re³Writer:** dear ms. [name], it was a pleasure taking care of you at [hospital]. you were admitted to the hospital with abdominal pain and found to have an infection in your gallbladder. you were treated with antibiotics for this infection. you also had a urinary tract infection which was treated with antibiotics. we made the following changes to your medications : - start oxycodone 5 mg every 6 hours as needed for pain.

Figure 3: An example of the PI generated by baseline and our approach (i.e., baseline with Re³Writer). Underlined text denotes alignment between the ground truth text and the generated text. Red colored text denotes unfavorable results. The Blue and Green colored text respectively denote the retrieved working experience and reasoned medical knowledge when generating corresponding sentences.

*every 6 hours as needed for pain*" (Green-colored text)), for the patient, who *was admitted to the hospital with abdominal pain*. The baseline can only generate the correct reason for the patient's admission, however, it also generates a serious wrong instruction (Red-colored text). It proves the effectiveness of our approach in retrieving associated working experience and reasoning associated medical knowledge to aid PI generation. Meanwhile, we note that our method produces interpretable refining weight values that may help understand the contribution of working experience and medical knowledge towards PI generation: when generating words similar to those that appear in the retrieved working experience (Blue-colored text) and reasoned medical knowledge (Green-colored text), the corresponding $\lambda_1$ and the $\lambda_2$ would be significantly larger than their means, $\bar{\lambda}_1$ and $\bar{\lambda}_2$, to efficiently refine the relevant information from working experience and medical knowledge, respectively.

## 5   Conclusion

We propose a new task of Patient Instruction (PI) generation which attempts to generate accurate and faithful PIs. To address this task, we present an effective approach Re³Writer: Retrieve, Reason, and Refine, which imitates the working patterns of physicians. The experiments on our built benchmark clinical dataset verify the effectiveness and the generalization capabilities of our approach. In particular, our approach not only consistently boosts the performance across all metrics for a wide range of baseline models with substantially different model structures, but also generates meaningful and desirable PIs regarded by clinicians. It shows that our approach has the potential to assist physicians and reduce their workload. In the future, it can be interesting to generate personalized PIs by taking into account the patient's cognitive status, health literacy, and other barriers to self-care.

**Limitations and Societal Impacts:**    The training of our approach relies on a large volume of existing PIs. The current model performance may be limited by the size of the built dataset. This might be alleviated in the future by using techniques such as knowledge distillations from publicly-available pre-trained models, e.g., ClinicalBERT [15]. Although our approach has the potential of alleviating the heavy workload of physicians, it is possible that some physicians directly give the generated PI to the patients or guardians without quality check. Also for less experienced physicians, they may not be able to correct the errors in the machine-generated PI. To best assist the physicians via our approach, it is required to add process control to avoid unintended usage.

## Acknowledgments

This work was supported in part by the National Institute for Health Research (NIHR) Oxford Biomedical Research Centre; an InnoHK Project at the Hong Kong Centre for Cerebro-cardiovascular Health Engineering; and the Pandemic Sciences Institute, University of Oxford, Oxford, UK. Fenglin Liu gratefully acknowledges funding from the Clarendon Fund and the Magdalen Graduate Scholarship.

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
