# Retrieve, Reason, and Refine: Appendix of Generating Accurate and Faithful Patient Instructions

**Fenglin Liu**[1][*], **Bang Yang**[2][*], **Chenyu You**[3], **Xian Wu**[4], **Shen Ge**[4], **Zhangdaihong Liu**[1,5]

**Xu Sun**[6][†], **Yang Yang**[7][†], **David A. Clifton**[1,5]

[1]Department of Engineering Science, University of Oxford    [2]School of ECE, Peking University
[3]Department of Electrical Engineering, Yale University    [4]Tencent JARVIS Lab, China
[5]Oxford-Suzhou Centre for Advanced Research, China
[6]MOE Key Lab of Computational Linguistics, School of Computer Science, Peking University
[7]School of Public Health, Shanghai Jiao Tong University School of Medicine, China
{fenglin.liu, david.clifton}@eng.ox.ac.uk, {bangyang, xusun}@pku.edu.cn
chenyu.you@yale.edu, jessie.liu@oxford-oscar.cn
{kevinxwu, shenge}@tencent.com, emma002@sjtu.edu.cn

## A   Medical Knowledge Graph

In our work, we construct an off-the-shelf medical knowledge graph $\mathcal{G} = (V, E)$ ($V = \{v_i\}_{i=1:N_{\text{KG}}} \in \mathbb{R}^{N_{\text{KG}} \times d}$ is a set of nodes and $E = \{e_{i,j}\}_{i,j=1:N_{\text{KG}}}$ is a set of edges), which models the domain-specific knowledge structure, to explore the medical knowledge. In implementation, we consider all clinical codes (including diagnose codes, medication codes, and procedure codes) during hospitalization as nodes, i.e., each clinical code corresponds to a node in the graph. The edge weights are calculated by the normalized co-occurrence of different nodes computed from training corpus. Figure 1 gives an illustration of the constructed medical knowledge graph. It is worth noting that more complex graph structures could be constructed by using more large-scale external medical textbooks. Therefore, our approach is not limited to the currently constructed graph and could provide a good basis for the future research of Patient Instruction generation.

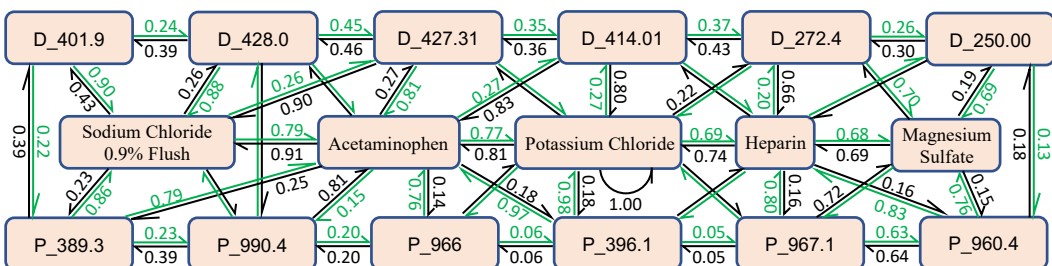

Figure 1: The constructed medical knowledge graph. Each clinical code corresponds to a node in the graph. We present the most frequent 6 diagnose nodes (the first row), 5 medication nodes (the second row), and 6 procedure nodes (the third row), and parts of their edge weights. Please refer to Table 1 for the exact meanings of these diagnose and procedure nodes.

---

[*]Equal contribution.
[†]Corresponding authors.

36th Conference on Neural Information Processing Systems (NeurIPS 2022).

Table 1: The exact meanings of the most frequent diagnose and procedure nodes in Figure 1.

| # | Diagnose Nodes | Procedure Nodes |
|---|---|---|
| 1 | **D_401.9**: Unspecified essential hypertension | **P_389.3**: Venous catheterization, not elsewhere classified |
| 2 | **D_428.0**: Congestive heart failure, unspecified | **P_990.4**: Transfusion of packed cells |
| 3 | **D_427.31**: Atrial fibrillation | **P_966**: Enteral infusion of concentrated nutritional substances |
| 4 | **D_414.01**: Coronary atherosclerosis of native coronary artery | **P_396.1**: Extracorporeal circulation auxiliary to open heart surgery |
| 5 | **D_272.4**: Other and unspecified hyperlipidemia | **P_967.1**: Continuous invasive mechanical ventilation for less than 96 consecutive hours |
| 6 | **D_250.00**: Diabetes mellitus without mention of complication, type II or unspecified type, not stated as uncontrolled | **P_960.4**: Insertion of endotracheal tube |

Table 2: Effect of the number of retrieved instructions $N_{\text{P}}$ in our Retrieve module when retrieving the working experience.

| $N_{\text{P}}$ | Dataset: Patient Instruction (PI) | | | | | | | |
|---|---|---|---|---|---|---|---|---|
| | METEOR | ROUGE-1 | ROUGE-2 | ROUGE-L | BLEU-1 | BLEU-2 | BLEU-3 | BLEU-4 |
| Baseline | 19.9 | 39.0 | 20.3 | 37.1 | 41.6 | 32.5 | 27.9 | 25.1 |
| 5 | 20.6 | 40.5 | **21.9** | 38.4 | 41.7 | 33.2 | 28.9 | 26.3 |
| 10 | 20.7 | 40.6 | **21.9** | 38.5 | 42.4 | 33.6 | 29.3 | 26.5 |
| 20 | **20.9** | **40.8** | **21.9** | **38.6** | **43.2** | **34.2** | **29.7** | **26.8** |
| 30 | 20.5 | 40.4 | 21.8 | 38.3 | 42.0 | 33.4 | 29.0 | 26.3 |
| 50 | 20.3 | 40.1 | 21.5 | 37.9 | 41.8 | 33.1 | 28.8 | 26.0 |

For the constructed knowledge graph, we use randomly initialized embeddings $H^{(0)} = \{v_1, v_2, \ldots, v_{N_{\text{KG}}}\} \in \mathbb{R}^{N_{\text{KG}} \times d}$ to represent all node features. To obtain the final medical knowledge $G_{\text{Pr}} = \{v'_1, v'_2, \ldots, v'_{N_{\text{KG}}}\} \in \mathbb{R}^{N_{\text{KG}} \times d}$, we adopt graph convolution layers [5, 4, 3] to encode the graph $\mathcal{G} = (V, E)$, which is defined as follows:

$$H^{(l+1)} = \text{ReLU}(\hat{A}\hat{D}^{-1}H^{(l)}W^{(l)} + b^{(l)}), \quad l \in [0, L-1] \tag{1}$$

where ReLU denotes the ReLU activation function, $\hat{A} = A + I$ is the adjacency matrix $A \in \mathbb{R}^{N_{\text{KG}} \times N_{\text{KG}}}$ of the graph $\mathcal{G}$ with added self-connections, $I \in \mathbb{R}^{N_{\text{KG}} \times N_{\text{KG}}}$ is the identity matrix, $\hat{D} \in \mathbb{R}^{N_{\text{KG}} \times N_{\text{KG}}}$ is the out-degree matrix where $D_{ii} = \sum_j A_{ij}$, $W^{(l)} \in \mathbb{R}^{d \times d}$ and $b^{(l)} \in \mathbb{R}^d$ are trainable parameters, and L is the number of layers. We empirically set $L = 1$ and regard $H^{(1)} = \{v'_1, v'_2, \ldots, v'_{N_{\text{KG}}}\} \in \mathbb{R}^{N_{\text{KG}} \times d}$ as the medical knowledge $G_{\text{Pr}} \in \mathbb{R}^{N_{\text{KG}} \times d}$ in our Re$^3$Writer.

## B  Effect of the Number of Retrieved Instructions

Table 2 shows that all variants with different number of retrieved instructions $N_{\text{P}}$ can consistently outperform the baseline model, which proves the effectiveness of our approach in retrieving the working experience to boost the Patient Instruction generation. In particular, when the number of retrieved instructions $N_{\text{P}}$ is 20, the model gets the highest performance, explaining the reason why the value of $N_{\text{P}}$ is set to 20 in our Re$^3$Writer. For other variants, we speculate that when $N_{\text{P}}$ is set to small values, the model will suffer from the inadequacy of information. When $N_{\text{P}}$ is set to large values, retrieving more patient instructions will bring more irrelevant noise to the model, impairing the performance.

## C  Retrieve with Age and Gender Information

We further incorporate the demographic/personal information, e.g. age and gender, into our approach to match patients in the Retrieve module. Specifically, to ensure an even distribution of the data, we divide the ages into three age-groups: Age < 55 (29.9%), 55 <= Age < 70 (30.5%), and Age >= 70 (39.7%). As a result, given a new male/female patient at 61 years old, we will match male/female patients in the age-group 55 <= Age < 70 in the training data to generate the PIs. The results are reported in Table 3. The results show that the incorporation of demographic/personal information can

Table 3: Performance of our approach incorporating Age and Gender information to match patients in the Retrieve module.

| Methods | Age+Gender | Dataset: Patient Instruction (PI) | | | | | | | |
|---|---|---|---|---|---|---|---|---|---|
| | | METEOR | ROUGE-1 | ROUGE-2 | ROUGE-L | BLEU-1 | BLEU-2 | BLEU-3 | BLEU-4 |
| Seq2Seq | - | 19.9 | 39.0 | 20.3 | 37.1 | 41.6 | 32.5 | 27.9 | 25.1 |
| with Re$^3$Writer | - | 20.9 | **40.8** | 21.9 | 38.6 | 43.2 | 34.2 | 29.7 | 26.8 |
| | $\surd$ | **21.0** | **40.8** | **22.0** | **38.7** | **43.5** | **34.5** | **29.9** | **27.1** |
| Transformer | - | 21.8 | 42.1 | 21.6 | 38.9 | 47.1 | 36.8 | 31.4 | 27.3 |
| with Re$^3$Writer | - | 23.7 | 45.8 | 24.4 | 42.2 | 52.4 | 41.2 | 35.0 | 30.5 |
| | $\surd$ | **24.1** | **46.1** | **24.6** | **42.5** | **52.9** | **41.6** | **35.3** | **30.8** |

indeed further boost the performance, which further prove our arguments and the effectiveness of our approach.

## D  Multi-Head Attention and Feed-Forward Network

Transformer [7] including a Multi-Head Attention (MHA) and a Feed-Forward Network (FFN) have achieved several state-of-the-art results on natural language generation.

The MHA consists of $n$ parallel heads and each head is defined as a scaled dot-product attention:

$$\text{Att}_i(X, Y) = \text{softmax}\left( \frac{X\text{W}_i^{\text{Q}}(Y\text{W}_i^{\text{K}})^T}{\sqrt{d_n}} \right) Y\text{W}_i^{\text{V}}$$

$$\text{MHA}(X, Y) = [\text{Att}_1(X, Y); \dots; \text{Att}_n(X, Y)]\text{W}^{\text{O}} \tag{2}$$

where $X \in \mathbb{R}^{l_x \times d}$ and $Y \in \mathbb{R}^{l_y \times d}$ represent the Query matrix and the Key/Value matrix, respectively; $\text{W}_i^{\text{Q}}, \text{W}_i^{\text{K}}, \text{W}_i^{\text{V}} \in \mathbb{R}^{d \times d_n}$ and $\text{W}^{\text{O}} \in \mathbb{R}^{d \times d}$ are training parameters, where $d_n = d/n$; $[\cdot; \cdot]$ denotes concatenation operation.

Following the MHA is the FFN, defined as follows:

$$\text{FFN}(x) = \max(0, x\text{W}_{\text{f}} + \text{b}_{\text{f}})\text{W}_{\text{ff}} + \text{b}_{\text{ff}} \tag{3}$$

where $\max(0, *)$ represents the ReLU activation function; $\text{W}_{\text{f}} \in \mathbb{R}^{d \times 4d}$ and $\text{W}_{\text{ff}} \in \mathbb{R}^{4d \times d}$ stand for learnable matrices; $\text{b}_{\text{f}}$ and $\text{b}_{\text{ff}}$ denote the bias terms. It is worth noting that both MHA and FFN are followed by an operation sequence of dropout [6], skip connection [2], and layer normalization [1].