# OpenReview forum: "Retrieve, Reason, and Refine: Generating Accurate and Faithful Patient Instructions"
_NeurIPS.cc/2022/Conference — NeurIPS 2022 Accept_

### Official Review · Reviewer_oy13 · 2022-07-11

**Rating:** 7
**Confidence:** 4
**Soundness:** 3 good
**Presentation:** 4 excellent
**Contribution:** 3 good

**Summary:**

The paper proposed the way (Re$^{3}$Writer) to automatically generate the Patient Instruction (PI) to reduce the workload of clinicians. Re$^{3}$Writer consists of three components: Retrieve (to retrieve instructions of previous patients according to the similarity of diagnosis, medication and procedure), Reason (to reason medical knowledge into the current input patient data with the help of the Knowledge Graph), and Refine (to refine relevant information from the retrieved working experience and reasoned medical knowledge to generate final PIs). The authors built a dataset PI which is based on the MIMIC 3 data and showed experimentally that with the help of their method performance of 6 different SOTA models can be substantially boosted across all metrics (BLEU-4, ROUGE-L, METEOR). They also showed the results from human evaluations to evaluate the approach for the clinical practice routine.

**Questions:**

1. Will authors publish the PI dataset? Or it will be enough to use the provided code and instructions to generate it?
2. Could authors provide more details about the dataset generation?
3. Does eICU[*] dataset have the text notes? If yes, will it be possible to include the results on this data as well?

*Tom J Pollard, Alistair EW Johnson, Jesse D Raffa, Leo A Celi, Roger G Mark, and Omar Badawi. The eICU collaborative research database, a freely available multi-center database for critical care research. Scientific data, 5(1):1–13, 2018.

**Limitations:**

Limitations were discussed.

**Strengths And Weaknesses:**

Strength:

1. Overall the paper is well organised, easy and enjoable to read. The contributions and motivation are clearly put. In my opinion this paper is relevant to the people from Machine Learning for Healthcare domain. The problem of generating the clear PI is very relevant for the clinicians. The attempt of medical reports (not instructions) generation were done before, but most of them used other modalities (i.e images). Here authors provided the way to utilise only the text knowledge.
2. Previous approaches are clearly explained and cited; experimental settings are well explained.
3. With the help of Re$^{3}$Writer the performance was significantly increased (Table 2). Ablation studies clearly shows the impact of each component in the final result (Table 5).

Weaknesses:
1. More details about the PI dataset will be helpful: i.e did authors use discharge summaries? Also about the patients notes - they are sparse in time of the patient stay in hospital (i.e nursing and physicians notes, radiology reports), did author just concatenate all these notes, or did they perform the selection somehow?
2. Additional ablation studies about what type of notes (nursing notes, physicians notes, radiology) contributes more in the Re$^{3}$Writer performance. However, it is a minor comment.
3. Some relevant citations are missing, for example "Auto-Encoding Knowledge Graph for Unsupervised Medical Report Generation", Fenglin Liu, Chenyu You, Xian Wu, Shen Ge, Sheng wang, Xu Sun, NeurIPS'21. This paper (and not only this) also utilises the Knowledge Graph structure in order to generate the medical reports. Not the same idea, but it will be helpful for reader to see the full overview.

---

> ### Author Response · Authors · 2022-08-02
> **Response to Reviewer oy13**
>
> Thanks for your helpful comments! If you have further concerns, please feel free to contact us.
>
> > **Q1**: Additional ablation studies about what type of notes (nursing notes, physician notes, radiology reports) contributes more in the ReWriter performance.
>
> **A1**: Thank you for pointing out a potential analysis point! We follow your advice to further conduct ablation studies to understand what type of notes (i.e., nursing notes, physician notes, radiology reports) contribute more to our approach. In detail, we remove the nursing notes/physician notes/radiology reports from the input to evaluate the performance of our approach. The results are as follows:
>
> |Methods|METEOR|ROUGE-1|ROUGE-2|ROUGE-L|BLEU-1|BLEU-2|BLEU-3|BLEU-4|
> |-|:-:|:-:|:-:|:-:|:-:|:-:|:-:|:-:|
> |Ours (Seq2Seq)|**20.9**|**40.8**|**21.9**|**38.6**|**43.2**|**34.2**|**29.7**|**26.8**|
> |&emsp; w/o Physician Notes|19.7|39.5|20.7|37.4|40.2|31.7|27.3|24.5|
> |&emsp; w/o Nursing Notes|20.3|39.9|21.6|37.9|41.3|32.9|28.6|26.0|
> |&emsp; w/o Radiology Reports|20.4|40.5|21.8|38.4|41.5|33.0|28.7|26.0|
> ||
> |Ours (Transformer)|**23.7**|**45.8**|**24.4**|**42.2**|**52.4**|**41.2**|**35.0**|**30.5**|
> |&emsp; w/o Physician Notes|21.8|42.1|22.8|39.9|49.6|39.3|33.4|28.8|
> |&emsp; w/o Nursing Notes|22.9|44.3|23.5|40.7|51.3|40.1|34.2|29.7|
> |&emsp; w/o Radiology Reports|23.4|45.6|24.3|42.1|52.0|40.5|34.6|30.1|
>
> As we can see, removing the Physician Notes significantly decreases the performance of our approach, indicating that the Physician Notes contribute more to the performance of patient instruction (PI) generation. We will include more analyses in our revision.
>
> > **Q2**: Will authors publish the PI dataset?
>
> **A2**: Yes! We promise to release the PI dataset and the codes, including the pre-processing codes, training codes and testing codes, which have been attached to the supplementary material, upon publication. Meanwhile, we promise to provide detailed instructions to guide the readers to 1) build and pre-process our PI dataset; 2) train the baseline model and our proposed model; and 3) generate the PIs for previously-unseen patients using trained models.
>
> > **Q3**: Could authors provide more details about the dataset generation?
>
> **A3**: Yes, we promise to provide detailed instructions to help readers to construct and pre-process the PI dataset in our released codes.
> Meanwhile, here are the details to construct our PI dataset:
>
> 1) For each patient in the MIMIC-III v1.4 resource, the dataset includes various patient’s health records during hospitalization, we directly concatenated all available health records during hospitalization as input for a patient, e.g., admission notes, nursing notes, and radiology reports. For example, if a patient only has admission notes, our model will just rely on the available admission notes to generate the PI (L221-L224).
>
> 2) In the MIMIC-III v1.4 resource, the patient instructions (i.e., discharge instructions) are included in discharge summaries. Therefore, given raw entries of MIMIC-III, we first used the discharge summaries to extract the patient instructions and excluded entries without patient instructions and those with less than five-word counts in patient instructions. The remaining 35,851 entries involve 28,029 unique patients, 35,851 hospital admissions, and 35,851 pairs of health records and patient instructions (L207-L209).
>
> 3) We further extracted the clinical codes, including diagnosis codes, ICD-9 medication codes, and ICD-9 procedure codes, of 35,851 hospital admissions from the MIMIC-III (L141-L142).
>
> 4) As a result, each entry in our built PI dataset is associated with a hospital admission ID, a patient ID, the clinical codes, the patient’s health records, and a patient instruction.
>
> 5) At last, we randomly split the dataset into 80%-10%-10% train-validation-test sets according to the patient ID. Therefore, there is no overlap of patients between train, validation and test sets (L210-L211).
>
> > **Q4**: Does eICU[*] dataset have the text notes?
>
> **A4**: Many thanks for recommending the eICU dataset. It's a nice dataset. We noticed that there are lots of useful notes, e.g., care documentation and care plans, in the eICU dataset. Therefore, it's possible to build a new task of automatic care plan generation, which aims to use care documentation as input to generate the care plans. In particular, our experiments on the PI dataset verify the effectiveness and the generalization capability of our proposed approach in generating clinical text given clinical notes as input. It implies that our approach has the potential to be applied to other clinical text generation tasks, e.g., the care plan generation task in the eICU dataset. We will follow the promising direction to adapt our approach to the eICU dataset in our next version.
>
> > **Q5**: Some relevant citations are missing.
>
> **A5**: Thanks for your suggestion. We will take your advice to cite and compare with more works in image-based medical report generation.

---

> > ### Comment · Reviewer_oy13 · 2022-08-09
> > **Response to Author Rebuttal**
> >
> > Thank you for considering my suggestions. I don't have any further concerns/questions.

---

> > > ### Author Response · Authors · 2022-08-10
> > > **Many thanks again for your review and feedback!**
> > >
> > > We thank the reviewer for acknowledging the response. We are genuinely happy that our response properly addresses fellow reviewers' concerns. We thank the reviewer again for the constructive feedback which have helped us improve our paper!

---

### Official Review · Reviewer_uXD1 · 2022-07-12

**Rating:** 7
**Confidence:** 5
**Soundness:** 3 good
**Presentation:** 4 excellent
**Contribution:** 3 good

**Summary:**

In this paper, the authors have proposed a new task of generating patient instructions (PIs) using the MIMIC dataset. The authors also proposed a strong SOTA model for the task Re3Writer which takes into consideration the historic PIs, clinical domain knowledge and refining of the final generated PI for a patient. The authors show that their method can improve the performance of a lot of baseline models by following the three techniques defined in the ReWriter framework.

**Questions:**

Weakness/Questions:
These are some things on which I would like to hear from the authors.
1. During train and test split, did the authors make sure that the same patient is not used in train and test. Even if the HADM (hospital admission ID) is different, a same patient can have similar PIs such as older patients can have same recommendations during multiple visits.
2. Did you consider UMLS for knowledge injection? Because the information of a new patient with a new ICD code that has not been used before in the train set would get ignored. Whereas UMLS can help with that information, if you can run an entityLinker to get different clinical entities and then create your final graph.
3. The patients were matches via their procedure, diagnostic codes but their other demographic/personal information such as age and gender wasn't considered. A patient with same ICD codes but with different age-group could potentially be assigned different PIs because of their health.
4. Note: I might be wrong here but why not consider a pre-trained model? Is that because of constraint of the note length?

**Limitations:**

Currently in this draft, no serious limitations are discussed. It is great to see Fig.4 where the model performs exceptionally well than the baseline model, it would be great to see some examples where ReWriter is consistently wrong.

**Strengths And Weaknesses:**

Strength of the paper:
1. It tackles an important problem and given it is created with the help of an openly available dataset (MIMIC), other researchers can also access the dataset and build further models.
2. The ReWriter approach is very intuitive as it considers the past PIs which generating the PI for a new patient and then considers the medical knowledge for refining it. This medical knowledge further helps the model in generating clinically coherent PIs.
3. All experiments are well defined but I really like the Table 5 ablation study because it justifies the use of each step in ReWriter.

Weakness/Questions:
These are some things on which I would like to hear from the authors.
1. During train and test split, did the authors make sure that the same patient is not used in train and test. Even if the HADM (hospital admission ID) is different, a same patient can have similar PIs such as older patients can have same recommendations during multiple visits.
2. Did you consider UMLS for knowledge injection? Because the information of a new patient with a new ICD code that has not been used before in the train set would get ignored. Whereas UMLS can help with that information, if you can run an entityLinker to get different clinical entities and then create your final graph.
3. The patients were matches via their procedure, diagnostic codes but their other demographic/personal information such as age and gender wasn't considered. A patient with same ICD codes but with different age-group could potentially be assigned different PIs because of their health.
4. Note: I might be wrong here but why not consider a pre-trained model? Is that because of constraint of the note length?

---

> ### Author Response · Authors · 2022-08-02
> **Response to Reviewer uXD1**
>
> Thanks for your helpful comments! If you have further concerns, please feel free to contact us.
>
> > **Q1**: During train and test split, did the authors make sure that the same patient is not used in train and test?
>
> **A1**: We have made sure that the same patient is not used in both training and testing. The statistics of our dataset in Table 1 show that the average hospitalization rate per patient is 28,673/22,423=1.28. As described in L210-L211, we randomly divided the dataset into train-validation-test partitions according to patient ID instead of hospital admission ID. Therefore, there is no overlap of patients between train, validation and test sets. The goal of our work is to generate patient instructions (PIs) for previously-unseen patients.
>
> > **Q2**: Did you consider UMLS for knowledge injection and creating the final graph?
>
> **A2**: Thank you for the suggestion. According to the ablation study in Table 5, the incorporation of a relatively small knowledge graph built on the internal training set has already brought 8% relative improvement in BLEU-4. We agree with your point that a more complex and larger graph structure, which is constructed by using larger-scale well-defined external medical ontologies/textbooks, e.g., UMLS, can further boost the performance and the generalization capability of the approach. And it is a good direction to use the UMLS to boost the performance and particularly process a new patient with an unseen new ICD code. We will explore it in future works.
>
> > **Q3**: The patients were matched via their procedure, diagnostic codes but their other demographic/personal information such as age and gender wasn't considered.
>
> **A3**: Thank you for pointing out a potential analysis point. We follow your advice to incorporate age and gender information into our approach to match patients. Specifically, to ensure an even distribution of the data, we divide the ages into three age-groups: Age < 55 (29.9%), 55 <= Age < 70 (30.5%), and Age >= 70 (39.7%). As a result, given a new male/female patient at 61 years old, we will match male/female patients in the age-group 55 <= Age < 70 in the training data to generate the PIs. The results are reported in the Table below.
>
> | Methods              | METEOR         | ROUGE-1 | ROUGE-2 | ROUGE-L | BLEU-1         | BLEU-2         | BLEU-3         | BLEU-4         |
> | :------------------- | :------------: |:------------: |:------------: |:------------: |:------------: |:------------: |:------------: |:------------: |
> | Seq2Seq [1]          | 19.9  |39.0  |20.3  |37.1  |41.6  |32.5  |27.9  |25.1  |
> | &emsp; with Re$^3$Writer  |20.9  |**40.8**  |21.9  |38.6   | 43.2   | 34.2  |29.7  | 26.8   |
> | &emsp;&emsp; with Age+Gender    | **21.0** | **40.8** | **22.0** | **38.7** | **43.5** | **34.5** | **29.9** | **27.1** |
> | | | | | | |
> | Transformer [43]     |   21.8 | 42.1 |21.6 | 38.9 |47.1 |36.8 | 31.4 |27.3|
> | &emsp; with Re$^3$Writer |  23.7 | 45.8 |24.4   | 42.2 | 52.4 | 41.2 | 35.0 | 30.5 |
> | &emsp;&emsp; with Age+Gender      |   **24.1**  | **46.1**  | **24.6**  | **42.5** | **52.9** | **41.6** | **35.3** | **30.8** |
>
> The results show that the incorporation of demographic/personal information can indeed further boost the performance, we will follow your constructive advice to conduct detailed analyses in our revision.
>
> > **Q4**: Why not consider a pre-trained model?
>
> **A4**: Thank you for the advice. For the encoder of our approach, we adopted the pre-trained model BERT to encode the information of matched (i.e., retrieved) patients. For the decoder of our approach, which is used for language generation, in our preliminary experiments, we attempted to adopt the pre-trained model T5 [R1] to generate the final PIs. The results of the pre-trained model T5 and the model trained from scratch are:
>
> | Methods | METEOR| ROUGE-1 | ROUGE-2 | ROUGE-L | BLEU-1| BLEU-2| BLEU-3| BLEU-4|
> | :------- | :------------: | :----------: | :----: | :----: |  :----: |  :----: |  :---------: | :---------: |
> | T5 [R1]  | 20.5    |  41.2  |  21.3 |  38.7   | 44.6           | 34.7           | 29.8           | 26.1           |
> | Ours    | **23.7** | **45.8** |  **24.4**   | **42.2** | **52.4** | **41.2** | **35.0** | **30.5** |
>
> As we can see, the pre-trained model T5 performs worse than our model trained from scratch. We speculate that the T5 was pre-trained on general texts. Therefore, it's necessary to fine-tune the T5 to well adapt to the generation of clinical text. However, considering the expensive computational resources consumed by the pre-trained model T5, we did not further fine-tune the pre-trained model to take T5 as our decoder.
>
> [R1] Exploring the limits of transfer learning with a unified text-to-text transformer. JMLR, 2020.

---

> > ### Comment · Reviewer_uXD1 · 2022-08-07
> > **Response to Author Rebuttal**
> >
> > Thank you for considering my suggestions! I appreciate the quick turnaround on results and it is good to see that considering demographic information does help in improving the model performance further. I don't have any further concerns/questions. Thank you.

---

> > > ### Author Response · Authors · 2022-08-10
> > > **Many thanks again for your review and feedback!**
> > >
> > > We thank the reviewer for acknowledging the response. We are genuinely happy that our response properly addresses fellow reviewers' concerns. We thank the reviewer again for the constructive feedback which have helped us improve our paper!

---

### Official Review · Reviewer_Wpt8 · 2022-07-12

**Rating:** 5
**Confidence:** 4
**Soundness:** 2 fair
**Presentation:** 3 good
**Contribution:** 2 fair

**Summary:**

This paper introduces a deep-learning approach called Re3Writer to imitate the clinicians’ working patterns for automatically generating patient instructions at the point of discharge from the hospital.

**Questions:**

The application was tested using one generated dataset. How did the authors analyse the accuracy of the generated PIs?
I would suggest performing the comparison of the output of the proposed deep-learning approach with more datasets/examples. Otherwise, it is hard to assess the novelty of the work.

Further technical explanations are needed (e.g., weights between nodes). For example, In Figure 4, under the reasoned medical knowledge box. What does 0.013 mean?

How the optimum model was selected? Further information regarding hyper-parameter tuning can be useful.

Building a PI dataset is defined as a novelty (contribution). How it is built and tested? Did the authors build the deep-learning approach on this dataset? If so, how can authors make sure its applicability to real-world health data (e.g., clinical validation)?

**Ethics Review Area:**

["I don’t know"]

**Limitations:**

The proposed approach is interesting. However, further technical details, validation and also the comparison of the deep-learning application on more datasets are needed to assess its novelty.

**Strengths And Weaknesses:**

Strengths:

1. Topic is timely
2. The application is interesting
3. The paper is easy to follow

Weaknesses:
1. Lack of technical details (e.g., hyper parameter tuning, information about node transitions/weights)
2. Lack of output validation
3. Novelty is limited

---

> ### Author Response · Authors · 2022-08-02
> **Response to Reviewer Wpt8 (Part 1)**
>
> Thanks for your helpful comments! If you have further concerns, please feel free to contact us.
>
> > **Q1**: How did the authors analyse the accuracy of the generated PIs?
>
> **A1**: We analysed the accuracy of the generated Patient Instructions (PIs) using both automatic metrics and human evaluation.
>
> For automatic evaluation, as shown in Table 2, we considered the widely-used natural language generation metrics, i.e., BLEU, ROUGE and METEOR, which are calculated based on the text matching between the generated PIs and referenced PIs annotated by professional clinicians. L238-L246 and Table 2 show that our approach can boost baselines consistently across all metrics, with relative improvements up to 20%, 11% and 19% in BLEU-4, ROUGE-L and METEOR, respectively.
>
> Meanwhile, as shown in Table 3 and Table 4 in Section 4.3, we invited three annotators, including an experienced clinician and two senior medical school students, to evaluate the generated PIs according to the quality and usefulness in clinical practice. All three annotators have sufficient medical knowledge. Specifically, L258-L264 and Table 3 evaluate the generated PIs from three perspectives: fluency, comprehensiveness and faithfulness. Table 4 shows that the proposed approach can generate more accurate PIs than the baselines, improving the usefulness of AI systems for better assisting physicians in clinical decision-making and reducing their workload.
>
> > **Q2**: The comparison of the output of the proposed deep-learning approach with more datasets/examples.
>
> **A2**: Thank you for pointing out a potential analysis point. In this paper, we evaluated the proposed approach with the MIMIC-III dataset which is real clinical data. We follow your constructive advice to evaluate the performance of our approach with more fine-grained datasets.
> Specifically, we further divide MIMIC-III into three sub-datasets according to **Age**, **Gender**, and **Disease**. Three tables below show the results of our approach Re$^3$Writer on the three sub-datasets:
>
> |Age Group||Age<55|||||55<=Age <70|||||Age>=70|||
> | :------ | :--------:  | :----------:  | :--------: |   :------:  |  :------:  |  :---------: | :--------: | :-------: | :------: | :----------: | :--------: |:-------: |:--------: |:--------: |
> | Methods| METEOR| ROUGE-L| BLEU-4|  |   | METEOR| ROUGE-L| BLEU-4|   |  | METEOR| ROUGE-L| BLEU-4|
> | Seq2Seq [1]| 18.2| 34.7| 21.9||| 20.7| 39.5| 26.7||| 20.7| 37.1| 26.5|
> | &emsp; with Re$^3$Writer | **19.2** | **35.6** | **23.7** || | **21.8** | **41.2** | **28.4** ||  | **21.5** | **38.9** | **28.1** |
> |||||||||||||||
> | Transformer [43] | 20.2| 36.9| 24.4|| | 23.1| 41.3| 28.5|  |  | 22.8| 39.0| 28.4           |
> | &emsp; with Re$^3$Writer | **22.9** | **40.1** | **28.5**  | | | **26.2** | **45.0** | **31.8** || |  **24.3** | **42.7** | **31.2** |
>
>
> |Gender|| Female|||||| Male ||
> | :--------- | :---------:  | :---------:  | :--------: | :--------: |   :------:  |  :------:  |  :--------------: | :------: | :--------: |
> | Methods| METEOR         | ROUGE-L        | BLEU-4              |  |  |    | METEOR         | ROUGE-L        | BLEU-4       |
> | Seq2Seq [1] | 19.8           | 35.9           | 25.0      |  |    |      | 20.0           | 38.0           | 25.2           |
> | &emsp; with Re$^3$Writer  | **20.6** | **37.6** | **26.3** |  |   |  | **21.1** | **39.5** | **27.2** |
> |||||||||||||
> | Transformer [43]              | 21.5           | 38.1           | 26.9           | |  |   | 22.0           | 39.6           | 27.6           |
> | &emsp; with Re$^3$Writer  | **23.2** | **41.3** | **30.1** |  |  |   | **24.1** | **43.0** | **30.8** |
>
>
> |Disease|| Hypertension||||||  Hyperlipidemia |||||| Anemia|||
> | :----------- | :----------:  | :--------:  | :--------: |   :------:  |  :------:  |  :-------: | :------: | :-------:  | :-----: | :----------: | :-----------: | :------: |:------: |:------: |:------: |:------: |
> | Methods| METEOR         | ROUGE-L        | BLEU-4              |||| METEOR         | ROUGE-L        | BLEU-4       |||| METEOR         | ROUGE-L        | BLEU-4|
> | Seq2Seq [1]| 21.3           | 39.8           | 27.9           |||| 21.3           | 41.7           | 27.2           |||| 18.0           | 36.4           | 20.7           |
> | &emsp; with Re$^3$Writer  | **22.6** | **41.4** | **30.4** |||| **22.5** | **43.9** | **29.5** |||| **18.8** | **37.6** | **22.3** |
> |||||||||||||
> | Transformer [43] | 22.8| 42.5| 30.7|||| 23.0| 44.7| 30.3 |||| 19.6| 38.2| 23.4|
> | &emsp; with Re$^3$Writer   | **24.6** | **45.1** | **33.5** |||| **24.9** | **46.4** | **33.8** |||| **21.8** | **41.3** | **27.4** |
>
> As we can see, the proposed approach can consistently boost baselines across different ages, genders and diseases on all evaluation metrics, proving the generalization capability and the effectiveness of our method to different datasets/examples. We will follow your advice to include more detailed analyses in our revision.

---

> > ### Author Response · Authors · 2022-08-02
> > **Response to Reviewer Wpt8 (Part 2)**
> >
> > > **Q3**: Further technical explanations are needed (e.g., weights between nodes).
> >
> > **A3**: The weights between nodes are calculated by normalizing the co-occurrence of pairs of nodes in the training corpus. In detail, as shown in L159-L160 and Appendix A in our Supplementary Material, we consider all clinical codes (including diagnosis codes, medication codes, and procedure codes) during hospitalization as nodes, i.e., each clinical code corresponds to a node in the graph. The weight between two nodes is calculated by the normalized co-occurrence of these two nodes counted from the training corpus. Figure 1 in the Appendix illustrates how the medical knowledge graph is built. In Figure 4, 0.013 means that the frequency of co-occurrence of "abdominal pain" and "urinary tract infection" is 0.013.
> >
> > > **Q4**: How the optimum model was selected?
> >
> > **A4**: We selected the optimum model based on the performances on the validation set. Specifically, for the hyper-parameter $N_\text{P}$ (the number of retrieved previous PIs), we reported the results and analyses of different $N_\text{P}$ in Appendix B of Supplementary Material. For the learning rate, we reported the effectiveness of our model in boosting the robustness of model to a wide range of learning rates in Figure 3. For the selected modules, we reported the performances and analyses of different modules in Table 5 and L280-L303 in our paper.
> >
> > > **Q5**: How it is built and tested? Did the authors build the deep-learning approach on this dataset? If so, how can authors make sure of its applicability to real-world health data (e.g., clinical validation)?
> >
> > **A5**: To build the dataset, as shown in L193-L199 of our paper, we collected and built the benchmark clinical dataset Patient Instruction (PI) based on the publicly-accessible MIMIC-III v1.4 resource (https://physionet.org/content/mimiciii/), which integrates de-identified, comprehensive clinical data for patients admitted to the Beth Israel Deaconess Medical Center in Boston, Massachusetts, USA. It means that the data we built was originally produced in real-world clinical settings, not synthetic or machine-generated.
> >
> > To test the dataset, in Section 4.2 of our paper, we incorporate the proposed approach into six representative language generation deep-learning models with different structures. As shown in Table 2 of our paper, our approach can boost the performance of baselines across all metrics. Meanwhile, in Section 4.3 of our paper, we further invited doctors to conduct the clinical validation to measure the effectiveness of our approach in terms of its usefulness for clinical practice. Table 3 and Table 4 show that our approach can generate meaningful and desirable PIs that are recognized by clinicians, indicating that our approach has the potential to assist physicians and reduce their workload.

---

> ### Author Response · Authors · 2022-08-10
> **Look forward to further feedback**
>
> Dear Reviewer Wpt8:
>
> Thanks again for all of your constructive suggestions, which have helped us improve the quality and clarity of the paper!
>
> Since the author-reviewer discussion period will end soon in 1 day, we appreciate it if you take the time to read our response and give us some feedback. Please don't hesitate to let us know if there are any additional clarifications or experiments that we can offer, we appreciate your suggestions. If our response resolves your concerns, we kindly ask you to consider raising the rating of our work.
>
> Many thanks for your time and efforts!
>
> Warmest regards,
>
> Authors

---

### Meta-Review · Area_Chair_ve9u · 2022-08-22

**Recommendation:** Accept
**Confidence:** Certain

**Metareview:**

The authors propose and evaluate a method to automatically generate "patient instruction" drafts. There was a consensus amongst reviewers that this is an interesting application.

While the technical innovation here may be modest, the empirical results firmly establish the benefits of the proposed "Re3Writer" approach. The ablations provided (both in the original submission and during author response) further strengthen the contribution.

Furthermore, the task definition and accompanying dataset (derived from MIMIC) constitute contributions which may lead to follow-up work. That said, in revised versions of the paper the authors should include additional details about the data and be explicit that they will release this (as mentioned by reviewer oy13).

**Award:**

No

---

### Decision · Program_Chairs · 2022-09-14

Accept